# Pharmacokinetics and Changes in Lipid Mediator Profiling after Consumption of Specialized Pro-Resolving Lipid-Mediator-Enriched Marine Oil in Healthy Subjects

**DOI:** 10.3390/ijms242216143

**Published:** 2023-11-09

**Authors:** Pilar Irún, Patricia Carrera-Lasfuentes, Marta Sánchez-Luengo, Úrsula Belio, María José Domper-Arnal, Gustavo A. Higuera, Malena Hawkins, Xavier de la Rosa, Angel Lanas

**Affiliations:** 1Centro de Investigación Biomédica en Red de Enfermedades Hepáticas y Digestivas (CIBEREHD), Instituto de Salud Carlos III (ISCIII), 50009 Zaragoza, Spain; pcarreralasfuentes@gmail.com (P.C.-L.); mjdompera@salud.aragon.es (M.J.D.-A.); alanas@unizar.es (A.L.); 2Instituto de Investigación Sanitaria Aragón (IIS Aragón), 50009 Zaragoza, Spain; msanchez@iisaragon.es; 3Centro Mixto de Investigación con Empresas (CEMINEM), Campus Rio Ebro, Universidad de Zaragoza, 50018 Zaragoza, Spain; ubelio@solutexcorp.com (Ú.B.); ghiguera@solutexcorp.com (G.A.H.); mhawkins@solutexcorp.com (M.H.); 4Faculty of Health Sciences, Campus Universitario Villanueva de Gállego, Universidad San Jorge, Villanueva de Gállego, 50830 Zaragoza, Spain; 5Service of Digestive Diseases, Hospital Clínico Universitario Lozano Blesa, 50009 Zaragoza, Spain; 6SOLUTEX GC, SL., 50180 Zaragoza, Spain; 7Departamento de Medicina, Psiquiatría y Dermatología, Facultad de Medicina, Campus Plaza San Francisco, Universidad de Zaragoza, 50009 Zaragoza, Spain

**Keywords:** lipids, omega-3, resolution, polyunsaturated fatty acids, specialized pro-resolving mediators, immune function, autoimmune disease, resolvins

## Abstract

Omega-3 polyunsaturated fatty acids (PUFAs) play a vital role in human health, well-being, and the management of inflammatory diseases. Insufficient intake of omega-3 is linked to disease development. Specialized pro-resolving mediators (SPMs) are derived from omega-3 PUFAs and expedite the resolution of inflammation. They fall into categories known as resolvins, maresins, protectins, and lipoxins. The actions of SPMs in the resolution of inflammation involve restricting neutrophil infiltration, facilitating the removal of apoptotic cells and cellular debris, promoting efferocytosis and phagocytosis, counteracting the production of pro-inflammatory molecules like chemokines and cytokines, and encouraging a pro-resolving macrophage phenotype. This is an experimental pilot study in which ten healthy subjects were enrolled and received a single dose of 6 g of an oral SPM-enriched marine oil emulsion. Peripheral blood was collected at baseline, 3, 6, 9, 12, and 24 h post-administration. Temporal increases in plasma and serum SPM levels were found by using LC-MS/MS lipid profiling. Additionally, we characterized the temporal increases in omega-3 levels and established fundamental pharmacokinetics in both aforementioned matrices. These findings provide substantial evidence of the time-dependent elevation of SPMs, reinforcing the notion that oral supplementation with SPM-enriched products represents a valuable source of essential bioactive SPMs.

## 1. Introduction

In 1971, Bang, Dyerberg, and Nielsen discovered the importance of omega-3 fatty acids in health and their benefits in inflammatory diseases by analyzing the plasma of 130 Eskimos from Umanak, a district on the northern part of the west coast in Greenland [1]. These essential nutrients must be taken from food sources since their production in mammals is limited. Over the years, interest in supplementation with omega-3 fatty acids has been proposed to counteract the hampers of inflammatory diseases. Therefore, it seems a good approach to supplement humans with these essential nutrients [2]. While most epidemiological evidence and randomized control trials (RCTs) indicate the protective role of omega-3 on general health and well-being, there are some clinical studies that have shown no effect [3]. Of note, the mechanisms of these protective effects are poorly understood. To this, Prof. Serhan discovered that termination of inflammation is an active process orchestrated by natural lipid mediators, namely specialized pro-resolving mediators (SPMs) [4,5,6,7,8].

Eicosapentaenoic acid (EPA) and docosahexaenoic acid (DHA) are needed nutrients in the human diet at all stages of life for the resolution of inflammation [9]. Free fatty acids (FFAs) EPA, DHA, arachidonic acid (AA), and docosapentaenoic acid (DPA) are the substrates required for enzymes to convert from omegas ultimately into SPMs. This conversion of FFA first to monohydroxylated mediators, also known as monohydroxylated SPMs, and then to di- and tri-hydroxo SPMs, follow complex synthetic pathways that involve epoxydated transient intermediates as well as a variety of enzymes (e.g., cyclooxygenases, lipoxygenases) [10,11,12]. SPMs are a family of lipid mediators classified as lipoxins (LXs), resolvins (Rvs), maresins (MaRs), and protectins (PDs). Lipoxins derive from AA, while D-series Rvs (RvDs), MaRs, and PDs derive from DHA, and E-series Rvs (RvEs) derive from EPA. SPMs carry out their actions via G-protein-coupled receptors (GPCR) receptors and have critical roles in cell signaling [13,14,15], and they have been involved in arthritis [16], atherosclerosis [17], diabetes, Alzheimer’s disease [18], sepsis [14], and inflammatory bowel disease, among others [19]. 

SPMs act by limiting neutrophilic infiltration to the inflammatory focus, promoting the clearance of apoptotic cells and cellular debris, enhancing efferocytosis, counter-regulating the production of pro-inflammatory mediators such as chemokines and cytokines, and promoting pro-resolving macrophage skewing. Another action of the SPMs is to instruct the adaptive immune system or guide tissue repair and regeneration processes [16,20]. Their bioactive concentrations have been established in different human organs, tissues, and fluids such as brain [21,22], vagus nerve [23], lymph node [24], spleen [24,25], adipose tissue [26], placenta [27], bone marrow [28], cerebral spinal fluid [29], synovial fluid [30], breast milk [31], urine [32], as well as plasma and serum [33]. These observations support the hypothesis that supplementing with omega-3 fatty acids will increase SPM amounts in circulation as well as in organs and tissues, and their protective actions are mediated via the upregulation of SPM biosynthesis. Recent studies have described that after supplementation of omega-3-rich oils, SPM amounts are elevated in plasma [34]. Nonetheless, the interrelationship between supplementation and increased concentrations of SPMs, their biosynthesis in organs, and reprogramming of the immune systems is not well understood yet.

Here, we aim to determine the interconnection between the intake of essential nutrients like SPM-enriched oil and how to increase concentrations of the bioactive lipid mediators in peripheral blood. We also intend to determine the pharmacokinetic fundamentals. In order to attain this, we assessed the temporal amounts of circulating SPMs in 10 healthy individuals after consumption of 100 mL of SPM-enriched emulsion. A water cycle was also included to control normal variations according to circadian rhythms. Results from this study demonstrated that supplementation with an SPM-enriched marine oil leads to an increase in peripheral blood SPM concentration, indicating a role for SPM in mediating the actions of this supplement.

## 2. Results

Ten healthy volunteers were recruited (five women and five men), all of whom completed the study; for demographic information, see Table 1. The average age was 31.81 ± 5.95 years, and the BMI mean was 23.94 ± 3.41 kg/m^2^. No participants reported any side effects after supplementation, and it was well-tolerated by all during the study. To study the time-dependent levels of SPMs in circulation after SPM-enriched marine oil supplementation, plasma, and serum at 0, 3, 6, 9, 12, and 24 h were submitted to targeted lipidomics profiling using LC-MS/MS (see the Materials and Methods section). 

To begin, we instituted a control group for comparative assessment, administering a 100 mL oral water dose at identical time intervals. After assessing lipid mediators profiling in plasma (Appendix A, Figure 1), we observed that volunteers with oral water dosing exhibited either negligible increments or, in certain instances, minor reductions in omega-3 FFA over the 24 h period commencing from baseline (time 0). Notably, a discernible reduction in (DPA was observed at the 9 h mark, while at the 12 h mark, a reduction in both EPA and DPA was exhibited. By contrast, after oral supplementation of SPM-enriched emulsion, we observed a statistically significant increase at almost all times for EPA and DPA compared to time 0 h. EPA was increased at all time points and remained elevated. It increased statistically significantly by 2.6-fold at all measured times, except at 6 h when it reached 3.3-fold. DPA was elevated at all time points except for 9 h. This FFA peaked at 6 h with an increase of 4.2-fold. Interestingly, concentrations of AA did not change as well as DHA. Parallel investigation of serum samples obtained from the same participants demonstrated a concordance in the relative abundance trends of FFAs in serum (Appendix A, Figure 1), akin to those observed in plasma. AA and DHA remained unaltered, while EPA and DPA were statistically significantly increased following SPM-enriched oil supplementation. It is worth mentioning that the DPA was increased only from 3 h to 6 h and peaked at the 6 h mark by a 2.5-fold rise. In addition, when evaluating serum samples subsequent to oral water dosing, no alterations in FFA concentrations were detected over time. Together, this suggests that profiling for FFAs, either in plasma or serum, was equally important.

We next focused on SPM metabolomes, which were profiled in both plasma and serum. The EPA metabolome included 18-hydroxyeicosapentaenoic acid (18-HEPE) and RvE1. For human plasmas after water ingestion, their profiles indicated that there were no changes in the metabolites analyzed. For example, 18-HEPE remained unaltered over time compared to time 0. However, interestingly, this metabolite was statistically significantly elevated after SPM-enriched oil supplementation at any time point from 3 to 24 h compared to time 0. In fact, it increased 134.1-fold times at 3 h and progressively reduced over time, evolving into 74.5-fold at 6 h, 12-fold at 9 h, 4.1-fold at 12 h, and 3.1-fold at 24 h after supplementation (Appendix A, Figure 2). Subsequently, we conducted a parallel analysis within the serum matrix, scrutinizing the EPA metabolome. For serum samples, the EPA metabolome analysis revealed that similar to plasma, there were no changes after oral drinking water, except for a marginal decline in 18-HEPE at 12 h. On the other hand, after SPM supplementation, we found a statistically significant increase in 18-HEPE, peaking at a substantial 10-fold times increase during the 3 to 6 h period. This elevated concentration gradually waned over time, eventually reverting to its baseline level. Further metabolites in the synthetic EPA pathway, such as RvE1, remained undetectable in both plasma or serum regardless of supplementation with SPM-enriched oil or given drinking water in these settings (Appendix A, Figure 2).

The DHA metabolome included 14- hydroxy docosahexaenoic acid (14-HDHA), 17- hydroxy docosahexaenoic acid (17-HDHA), D-series Rv (RvD1 to RvD5), MaR1, MaR2, PDX, and PD1. In human plasmas after water ingestion, overall metabolite levels were not altered over time compared to time 0, although there was an increase in one metabolite (PDX) for one time point (3 h). By contrast, subsequent to SPM-enriched supplementation, we found a statistically significant increase in median values of several DHA-derived SPMs (e.g., RvD5, PDX, PD1, MaR1, MaR2, 14-HDHA, and 17-HDHA). The time-course progression relative to the levels of these SPMs was as follows: for RvD5, an 11.8-fold increase at 3 h and a 4.8-fold increase at 6 h. For PDX, a 16.5-fold increase at 3 h and a 6.7-fold increase at 6 h. For PD1, a 3.7-fold increase at 3 h and a 2.6-fold increase at 6 h. For MaR1, a 2.0-fold increase at 3 h. Notably, MaR2, with a basal median value of 0.0 pg/mL, exhibited a significant increase upon supplementation, reaching amounts of 39.34, 23.08, 3.23, 2.81, and 1.74 pg/mL at 3, 6, 9, 12, and 24 h, respectively. For 17-HDHA, it demonstrated a 38.0-fold increase from 3 to 6 h, followed by 4.3-fold and 1.7-fold at 9 and 12 h, respectively. For 14-HDHA, remarkable increases of 24.9-fold and 45.9-fold were observed at 3 h and 6 h, respectively. (Appendix A, Figure 3). Interestingly, we found an increase in RvD1 amounts at 6 h with a 3.6-fold. Also, RvD3 was increased by 2.2-fold times at 9 h post-supplementation. These findings suggest that following oral ingestion of ω-3 enriched with SPMs, as early as 3 h from the ingestion, the levels of SPMs in circulating blood increased, particularly for RvD5, PDX, PD1, MaR1, MaR2, 14-HDHA, and 17-HDHA and later at 6 h for RvD1 and at 9 h for RvD3.

Subsequently, we extended our investigation to the same DHA metabolome, this time analyzing serum samples (Appendix A, Figure 3). In human serum following oral water ingestion, we observed only minor variations in metabolite levels when compared to the baseline (time 0), primarily characterized by reductions in 17-HDHA, PD1, and PDX at 12 h, as well as a decline in 14-HDHA at 9 and 12 h, with the exception of increased MaR1 levels at 3 h and 9 h. By contrast, following SPM supplementation, similar to our findings in plasma, we detected a statistically significant median increase in several DHA-derived SPMs (e.g., RvD5, PDX, PD1, MaR2, 14-HDHA, and 17-HDHA). The time-course progressions from baseline median levels of these SPMs were as follows: RvD5 showed an approximate 3.6-fold increase at 3 and 6 h. For PD1, a 1.6-fold increase at 3 h, while at 6 h, an increase of 2.9-fold for PDX, 2.2-fold for 14-HDHA, and 4.5-fold for 17-HDHA. That is to say, after oral supplementation, serum SPM levels exhibited increases as early as 3 h for RvD5 and PD1 and at 6 h for PDX, MaR2, 14-HDHA, and 17-HDHA.

The AA metabolome included pro-resolving lipoxins (LXA_4_ and LXB_4_) and pro-inflammatory derived eicosanoids (PGE_2_, PGD_2_, PGF_2α_, LTB4 and TXB_2_). In human plasmas following water ingestion, no significant alterations were noted in the analytes studied, with the exception of LTB4, which exhibited an unexpected increase at 9 h and 24 h and a slight decrease in PGE_2_ at 12 h. After emulsion supplementation, all analyzed metabolites remained largely unaltered, except for PGE_2_, which augmented from 3 to 6 h, and TXB_2_ at a single time point increase at 3 h. On the other hand, when examining the AA metabolome in serum samples without supplementation, no changes over time were observed. However, upon supplementation with the SPM emulsion, we observed a reduction in PGD_2_ levels at 6 h and a decline in LXA_4_ levels at 9–12 h after supplementation (Appendix A, Figure 4).

After observing increased pro-resolving mediators in circulation after a single dose of supplementation, we conducted a more in-depth statistical analysis. We employed a partial least squares discriminant analysis (PSL-DA) regression model rooted in the differential expression of lipid mediator concentrations to compare the non-supplemented phase (oral drinking water) and the phase with supplementation of omega-3 enriched with SPMs. This analysis aimed to pinpoint the key lipid mediators upregulated after dosing. In plasma, the variable selection outputs at 3–6 h post-supplementation show differential increases in PDX, PD1, RvD5, MaR2, 14-HDHA, 17-HDHA, and 18-HEPE, along with omega-3 DPA and EPA. Additionally, at 6 h post-supplementation, RvD1 and RvD2 were also differentially increased in the treatment group (Figure 5). Comparatively, in serum, the variable selection outputs at 3–6 h post-supplementation showed a similar pattern to that observed in plasma. Notably, PDX, PD1, RvD5, MaR2, 14-HDHA, 17-HDHA, and 18-HEPE, together with omega-3 DPA and EPA, demonstrated significant differential increases in the treatment group (Figure 6). In conclusion, this analysis indicates that aside from EPA, 18-HEPE, 14-HDHA, and 17-HDHA, which are included in the emulsion, PDX, PD1, MaR2, RvD5, and DPA were the major contributors to the separation between the supplemented and non-supplemented phases.

Having established the concentrations over time subsequent to the oral supplementation of SPM-enriched dose in peripheral blood, we assessed the pharmacokinetic parameters (e.g., maximum concentration (Cmax), time to reach Cmax (Tmax), and the area under the concentration–time curve over 24 h (AUC (0–24 h))) for EPA, DHA, 14-HDHA, 17-HDHA, and 18-HEPE along with their derived SPMs in both plasma and serum. Notably, a single dose of the SPM-enriched marine oil administered to healthy subjects resulted in significant increases, when compared to the control group receiving water dose, of plasma Cmax and AUC (0–24 h) for EPA, 18-HEPE, 14-HDHA, 17-HDHA, RvD5, MaR2, and DPA. In addition, a decrease in Tmax was observed for PD1 and 14-HDHA and increased AUC (0–24 h) for PDX (Table 2). Relative to serum determinations, like that described in plasma, supplementation increased significantly compared to the control phase for both serum Cmax and AUC (0–24 h) of EPA, 18-HEPE, 14-HDHA, 17-HDHA, RvD5, MaR2, and DPA. In contrast to that observed in plasma, increased Cmax and AUC (0–24 h) for PD1 and increased not only AUC (0–24 h) but also Cmax and decreased Tmax were observed for PDX. Furthermore, changes in serum parameters were observed for certain AA-derived metabolites, including reductions in Cmax and AUC (0–24 h) for PGD_2_ and reductions in Tmax and AUC (0–24 h) for PDF_2α_. A reduction in Tmax was also observed for 18-HEPE (Table 3).

## 3. Discussion

Omega-3s play a pivotal role in health, well-being, and inflammatory diseases in humans. When adequately acquired in the diet, they protect individuals from excessive inflammation and counteract internal inflammatory mediators to maintain homeostasis, in part via actions carried by the bioactive omega-3 further metabolites like SPMs. The origin of resolvins, maresins, lipoxins, protectins, and monohydroxylated SPMs in plasma and serum has yet to be identified. It is possible that they originate in organs that produce SPMs, such as bone marrow [35], adipose tissue [36], spleen [37], and brain [38], or are produced through local thrombotic events [39], which are later distributed throughout the circulation. Further clinical studies need to be performed in order to establish their full potential as pharmaceutical agents for resolving inflammation [4]. Meanwhile, supplementing with them appears to be a good strategy to increase their circulating levels. Several endogenous mechanisms for de novo synthesis of SPMs have been elucidated. For example, one method to increase sufficient levels of omega-3 is through transgenic mice that express the FAT-1 gene, which increases the synthesis of RvE1, RvD3, and PD1. These animals presented protection in an inflammatory model of colitis [40]. Additionally, certain gases have been reported to elevate SPM levels. In fact, inhalation of carbon monoxide accelerates resolution via the SPM–heme oxygenase-1 axis [41]. Also, AT-LXA_4_, produced after a low dose of aspirin, synthesizes nitric oxide (NO), which promotes resolution by reducing leukocyte addition and limiting infiltration [42]. Moreover, aspirin triggers the synthesis of lipid metabolites that directly halt neutrophil trafficking and rattle pro-resolution actions [43]. RvD2 triggers NO production in human umbilical vein endothelial cells (HUVECs) and promotes resolution, reducing neutrophil trafficking [44]. Recently, in an animal model of neurodegenerative disease, it was shown that sphingosine kinase1 (SphK1) acetylates COX-2, increasing endogenous production of AT-LXA_4_ and RvE1, resulting in reduced pathology by enhancing phagocytosis of the microglial cells [45]. In addition, some drugs also increase SPM amounts, such as aspirin [46], statins, and dexamethasone [47,48,49]. The increase in SPMs triggered by drugs can also be attributed, in part, to endogenous mechanisms. Their cumulative production, both endogenously and through supplementation or drug-induced pathways, would establish a homeostatic and balanced scenario [4]. Together, all these strategies to increase SPMs offer a rationale for SPM therapy to promote the resolution of inflammation.

In humans, prenatal omega-3 supplementation increases the concentration of 18-HEPE and 17-HDHA in the placenta and umbilical blood, which might support immune functions [50]. Human breast milk also contains SPMs, and when infection (mastitis) occurs, it presents a distinct metabolipidomics profile skewed towards proinflammatory mediators [31]. All of these factors may contribute to the newborn immune system, microflora, and the resolution after delivery. Bioactive concentrations have been reported in human lymphoid organs such as the spleen and lymph nodes, as well as in plasma and serum after omega-3 supplementation.

After resistance exercise, recovery in healthy humans correlates with elevated serum SPMs, and, more interestingly, this is blocked by pretreatment with ibuprofen, a known NSAID [51]. Furthermore, a recent clinical randomized trial involving healthy subjects exposed to intense exercise concluded that exercise led to an increase in SPM concentrations in circulation [52]. Another double-blinded randomized clinical study reported that after a low dose of aspirin, epimer lipid mediators in the periapical fluid of necrotic teeth increased [53]. After an omega-3-enhanced diet, for people who suffer from migraines, pain reduction coincides with the intervention that reduces omega-6 and increases omega-3 in the diet. Interestingly, the elevated mediators in plasma were found to be monohydroxylated SPMs, specifically 17-HDHA, 18-HEPE, and tri-hydroxylated RvD2 [54]. Patients with sepsis in the ICU showed that their peripheral blood SPM-lipid-mediator profiles correlate with survival and acute respiratory distress syndrome development, thus suggesting plausible biologic targets for critical illness [55]. RvD3 was reduced in the serum of rheumatoid arthritis patients [56]. Furthermore, in line with the increased levels of SPMs in circulation, a recent randomized clinical study by Möller et al. revealed that oral supplementation of marine oils led to reduced pain in individuals with symptomatic knee arthritis when compared to a placebo group [57]. Also, RvD1 was significantly reduced in vulnerable regions of atherosclerotic plaques [58]. Visceral fat tissue from obese patients presented an imbalance of pro-resolving mediators versus pro-inflammatory mediators. Interestingly, the omental fatty tissue from these obese patients did not have the capacity to produce the same amounts of SPMs, denoting a lack of functional synthetic pathways for SPMs, although omega-3 FFAs were present [59].

It is now well appreciated how important it is to treat impaired resolution and regulate inflammation [4,60]. There is also growing evidence that supplementation with SPMs is a valid source of these active lipid mediators that will increase circulation to reset or reprogram the immune system [34]. In another study, it was observed that oral supplementation of SPM-enriched oil improved quality of life, reduced pain intensity or interference, and improved mood within 4 weeks of intervention in adults with chronic pain [61], and it was recently reviewed by Ji R.R. [62]. Schaller found that short-term supplementation dramatically remodels this synthetic pro-resolving pathway and induces a less inflammatory phenotype in circulating leukocytes as well [63]. In a clinical trial led by Prof. S.R. Shaikh, it was described that obese patients increased plasma levels of SPMs after supplementation, but interestingly, it modified their adaptive immune system [64]. So, an interesting next step would be to investigate whether patients with a variety of inflammatory diseases have similar or dysregulated temporal synthesis of SPMs and whether supplementation with SPM-enriched oils can reprogram this synthesis. 

Here, we provide evidence that after a single dose of 6 g of SPM-enriched marine oil supplementation, SPM concentrations increase in the peripheral blood (plasma and serum) of healthy subjects. Moreover, we provide evidence of the pharmacokinetics parameters of these mediators. Whether these increases in circulating SPMs are synthesized from endogenous mechanisms or absorbed after supplementation remains to be determined in further studies. Nevertheless, given that SPM amounts or synthesis of them are altered in disease, it seems that oral supplementation is a good source of essential SPMs. In this line, the deficient production of SPMs in obese patients due to the dysfunctional enzymatic activity of 15-LOX was restored by incubation with 17-HDHA, indicating that SPM supplementation with products similar to this study may be more efficient than using only their ω-3 precursors for supplementation [36]. Further studies are needed to evaluate the clinical potential of this SPM-enriched marine oil emulsion in diseases characterized by some degree of inflammation and/or impaired ability to synthesize SPMs.

## 4. Materials and Methods

### 4.1. Study Participants

Participants were healthy, non-smoking or without the use of nicotine-containing products within the last 6 months, men and women, 18 to 50 years, with a body mass index between 20 and 29 kg/m^2^, having good health according to medical history, vital signs, and routine laboratory test (biochemistry, hematology, and coagulation analysis). Subjects declared to be in good health and not to take any dexamethasone or statins. Exclusion criteria were the following: allergy to ω-3 acid or fish; positive screen for SARS-CoV-2 infection; consumption of xanthine and/or caffeine-containing products, alcohol, or vitamin supplements within 24 h; grapefruit and poppy-containing foods within 48 h; use of anti-inflammatory drugs within 7 days; consumption of fish oil supplements within 1 month; pregnant or breastfeeding women or those who have used contraceptive treatment within 3 months prior to enrolling; and subjects taking treatments that can interfere with the absorption, distribution, metabolism, or excretion of the product under research. Samples from 10 healthy subjects were included. All participants signed an informed consent document before any study procedures were performed. 

### 4.2. Study Design

All participants (*n* = 10) underwent a first blood extraction (plasma and serum) after a fasting period of at least 10 h prior to dosing, which was established as time 0 (t = 0). The times when participants received the meals (relative to the blood draw) were the following: For breakfast, immediately after 0 h blood collection; for lunch, immediately after 6 h blood collection; for snack, immediately after 9 h blood collection; and for dinner, immediately after 12 h blood collection. Then, as a basal value for each individual, participants were given 100 mL of water at breakfast time, and subsequent phlebotomy extractions were performed at 3, 6, 9, 12, and 24 h. At least 7 days later, the same phlebotomy protocol was assessed, but this time, the same participants received a single oral dose of 100 mL of the experimental supplementation emulsion instead of water. This emulsion was an SPM-enriched marine oil that contained 1556 mg of EPA-FFA, 3441 mg of DHA-FFA, 1.45 mg of 17-HDHA, 1.68 mg of 14-HDHA, 2.88 mg of 18-HEPE (in summation, 5981 mg total). Intervention compliance was assured by supervised administration of the control or intervention products by qualified study personnel of the research group. Standardized meals were provided during the 24 h extraction periods. A description of the diet is shown in Table 4. 

### 4.3. Safety Analysis

Participants were asked to report any side effects suffered after the supplement ingestion during the study days and when they returned for the 24 h sampling.

### 4.4. Biological Samples Processing

Samples were collected and processed as described here [65]. In brief, Peripheral blood samples were collected in 10 U/mL heparin and immediately centrifuged at 120 g for 20 min without braking to ensure that layers would not be disrupted. Then, plasma was transferred to 1.5 mL tubes, topped with a nitrogen gas, capped, and immediately stored at −80 °C for further processing. Serum samples were generated as described earlier by Norris et al. [39]. In brief, they were incubated overnight at 37 °C, allowing coagulation to occur, and then separated by centrifugation with the same protocol. 

### 4.5. Lipid Mediator Extraction and Profiling (LC-MS/MS)

Lipid mediators (LMs) were extracted from human serum samples according to a previously described solid-phase extraction (SPE) method [66]. In brief, each sample (plasma and serum, 1 mL) stored at −80 °C was thawed on ice. Internal labeled standards containing d8-5-HETE, d5-RvD2, d5-LXA_4_, d4-LTB4, and d4-PGE_2_ (500 pg each, Cayman Chemical Company) in 4 mL of methanol (Methanol Optima LC/MS Grade, Fisher Chemical) were added to each sample. Known concentrations of LMs in labeled standards were used for quantification purposes and posterior calculations on the recovery of LMs during the extraction process. Then, the samples were placed at −80 °C for 30 min for protein precipitation. Next, they were centrifuged at 890 g, 10 min, 4 °C. SPE columns were conditioned with 4 mL methanol and 4 mL water. Furthermore, Supernatants were quickly acidified to pH = 3.5 with 9 mL of acidic water (HCl) just prior to loading onto conditioned-SPE columns (100 mg, 10 mL, Biotage) and pH neutralized with 4 mL of MilliQ water, followed by a 4 mL of n-hexane wash step. After that, compounds were eluted with 9 mL of methyl format. Extracts from the SPE were dried under a stream of nitrogen and immediately after were resuspended in 50 μL methanol/water (50:50 vol/vol) (MeOH/Water Optima LC/MS Grade, Fisher Chemical, both) before injection into the LC-MS/MS system. For acquisition parameters, see below.

To determine the pharmacokinetic parameters and the effect of the concentrated emulsion of ω-3 PUFAs and their monohydroxylated metabolites on the production of pro-inflammatory and pro-resolving lipid mediators (LMs), plasma concentrations of 24 variables were quantified by LC-MS/MS. Each metabolome was analyzed in plasma and serum. The EPA metabolome included the monohydroxylated intermediate 18-HEPE and the resolvin RvE1, whereas the DHA metabolome included their monohydroxylated LMs (17-HDHA and 14-HDHA), D-series Rv (RvD1, RvD2, RvD3, RvD4, and RvD5), protectins (PD1 and PDX) and maresins (MaR1, MaR2). For the AA metabolome, the bioactive lipids included both pro-resolving lipoxins (LXA_4_, LXB_4_) and pro-inflammatory derived eicosanoids, including prostaglandins (PGE_2_, PGD_2_, and PGF_2α_), leukotriene B4 (LTB4), and thromboxane B_2_ (TXB_2_).

### 4.6. Targeted LC-MS/MS Acquisition Parameters

LC-MS/MS system consisted of a Qtrap 5500 (Sciex) equipped with a Shimadzu LC-20AD HPLC. A Kinetex Core–Shell LC-18 column (100 mm × 4.6 mm × 2.6 μm, Phenomenex) was housed in a column oven maintained at 50 °C. A binary eluent system of LC-MS/MS grade water (A) (Fisher Chemical) and LC-MS/MS grade methanol (Fisher Chemical) (B), both with 0.01% (*v*/*v*) of acetic acid, were used as mobile phase. LMs were eluted in a gradient program with respect to the composition of B as follows: 0–2 min, 50%; 2–14.5 min, 80%; 14.6–25.0 min; 98%. The flow rate was 0.5 mL/min. 

The QTRAP 5500 was operated in negative ionization mode, using scheduled multiple-reaction monitoring (MRM) coupled with the information-dependent acquisition (IDA) and an enhanced product ion scan (EPI). Each LM parameter (collision energy, target retention time, and specific first and third quadrupole mass transitions) was optimized according to reported methods [28]. For monitoring and quantification purposes, the amounts of LMs of interest were estimated as area under the peak, specifically using MRM with MS/MS matching signature retention time for each molecule with standards (<0.1 picograms was considered below the limit of detection). The lower limits of quantification (LLOQs) were determined by analyzing serial dilutions of the lower calibrator as the concentrations of each SPM with a signal/noise ratio ≥ 5-fold the signal/noise ratio of a blank solution according to the guidelines of the US Food and Drug Administration [67] and are listed in Table 5. Each molecule was validated using MRM with MS/MS matching signature ion fragments for each molecule (at least six diagnostic ions). The laboratory analyses were performed by Solutex GC SL in the CEMINEM, Universidad de Zaragoza, Zaragoza, Spain.

### 4.7. Pharmacokinetic Analysis

Pharmacokinetic parameters, including maximum concentration (Cmax), time to maximum concentration (Tmax), and area under the curve from time 0 to 24 h (AUC_0–24 h_) for the 24 variables previously mentioned, were analyzed after baseline-adjusting the concentration of each analyte for each participant as the concentration at a determined time minus that at time 0 (t = 0).

### 4.8. Statistical Analyses

An initial exploratory analysis of all clinical variables was carried out. Continuous variables were expressed as means with standard deviation, whereas qualitative variables were expressed as frequencies. Time and intervention evaluations on lipid mediators were performed. For the time effect, the significance of the change in values at hours 3, 6, 9, 12, and 24 from baseline (t = 0) within each situation was determined by Wilcoxon signed-rank test. Time and intervention evaluations of lipid mediators were performed. Partial least squares discriminant analysis (PLS-DA) was also performed by the PKNA R package. The mixOmics R package was used for graphical functions. Statistical significance was considered as *p*-value < 0.05. All statistical analyses were conducted with the R programming code (the R Foundation for Statistical Computing, Vienna, Austria).

## Figures and Tables

**Figure 1 ijms-24-16143-f001:**
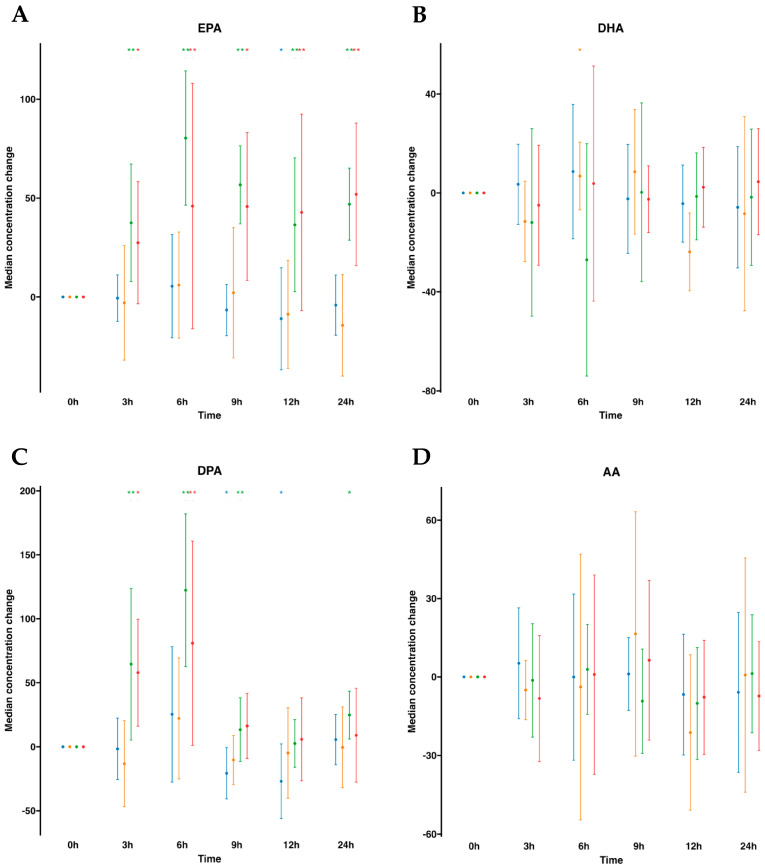
Median changes in plasma and serum free fatty acids. Mean changes in FFA concentrations at 3, 6, 9, 12, and 24 h relative to baseline after water or SPM-enriched marine oil intake for EPA (**A**), DHA (**B**), DPA (**C**), and AA (**D**). Values are expressed in ng/mL. Diagrams represent control plasma (
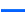
), supplemented plasma (

), control serum (
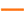
), and supplemented serum (
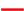
). Statistical significance for the time effect was determined by the Wilcoxon signed-rank test (* *p* < 0.05; ** *p* < 0.01).

**Figure 2 ijms-24-16143-f002:**
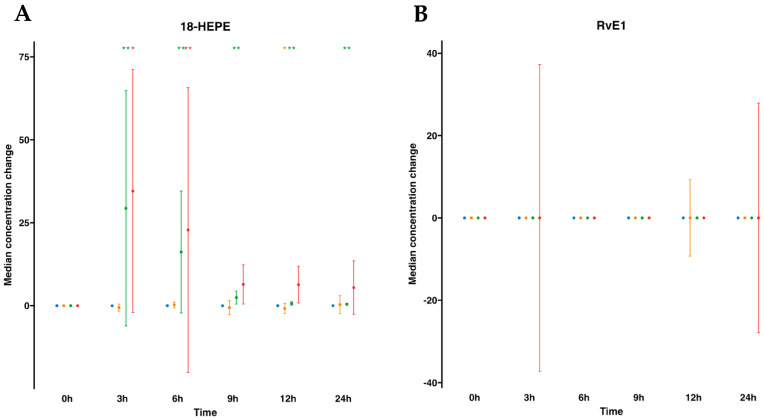
Median changes in plasma- and serum-EPA-derived lipid mediators. Mean changes in 18-HEPE (**A**) and RvE1 (**B**) concentrations at 3, 6, 9, 12, and 24 h relative to baseline after water or SPM-enriched marine oil intake. Values are expressed in ng/mL for 18-HEPE and pg/mL for RvE1. Diagrams represent control plasma (
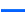
), supplemented plasma (

), control serum (
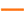
), and supplemented serum (
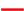
). Statistical significance for the time effect was determined by the Wilcoxon signed-rank test (* *p* < 0.05; ** *p* < 0.01).

**Figure 3 ijms-24-16143-f003:**
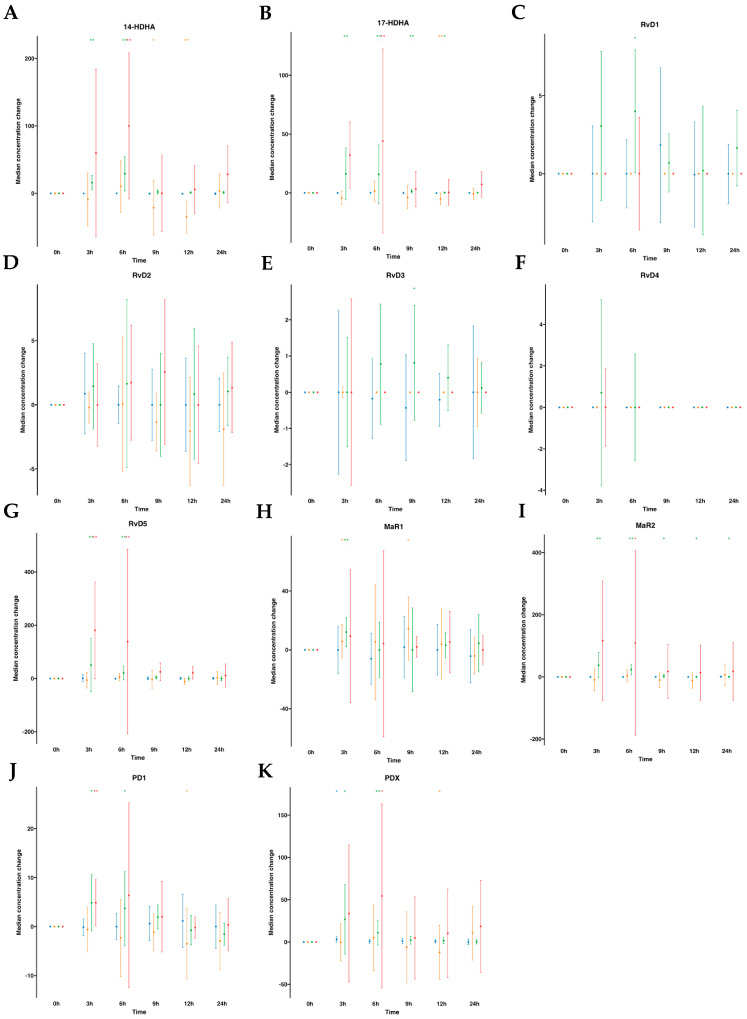
Median changes in plasma- and serum-DHA-derived lipid mediators. Mean changes in DHA-derived lipid mediators’ concentrations at 3, 6, 9, 12, and 24 h relative to baseline after water or SPM-enriched marine oil intake. Values are expressed in pg/mL except for 14/17-HDHA, which are expressed in ng/mL. Diagrams represent control plasma (
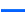
), supplemented plasma (

), control serum (
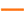
), and supplemented serum (
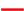
). Statistical significance for the time effect was determined by Wilcoxon signed-rank test (* *p* < 0.05; ** *p* < 0.01). Results in multipaneled figure refers to 14-HDHA (**A**), 17-HDHA (**B**), RvD1 (**C**), RvD2 (**D**), RvD3 (**E**), RvD4 (**F**), RvD5 (**G**), MaR1 (**H**), MaR2 (**I**), PD1 (**J**), and PDX (**K**), respectively.

**Figure 4 ijms-24-16143-f004:**
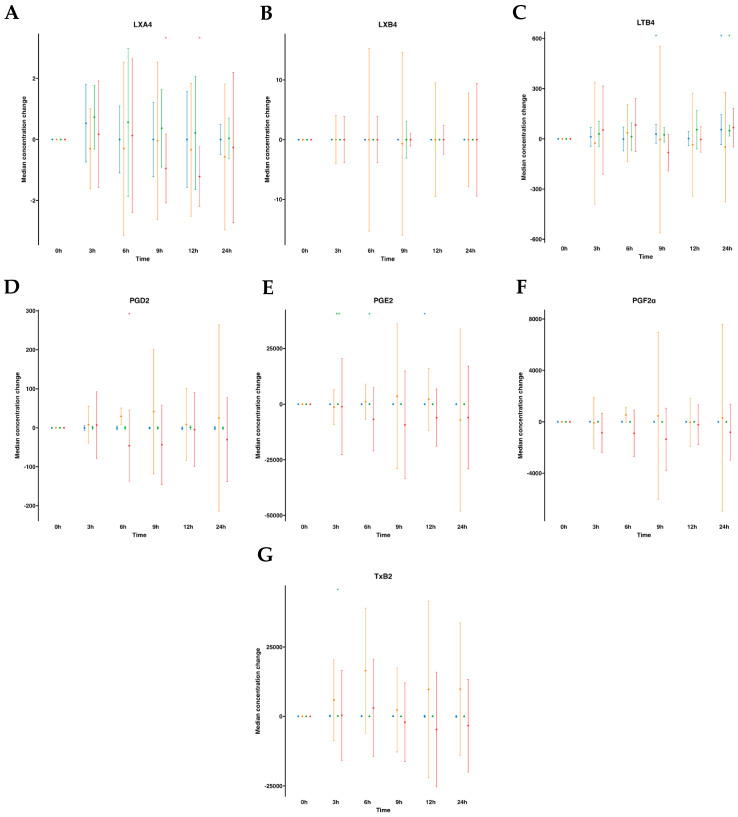
Median changes in plasma- and serum-AA-derived lipid mediators. Mean changes in LXA_4_ (**A**), LXB_4_ (**B**), LTB4 (**C**), PGD_2_ (**D**), PGE_2_ (**E**), PGF2α (**F**), and TXB_2_ (**G**) concentrations at 3, 6, 9, 12, and 24 h relative to baseline after water or SPM-enriched marine oil intake. Values are expressed in pg/mL. Diagrams represent control plasma (
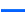
), supplemented plasma (

), control serum (
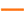
), and supplemented serum (
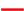
). Statistical significance for the time effect was determined by the Wilcoxon signed-rank test (* *p* < 0.05; ** *p* < 0.01).

**Figure 5 ijms-24-16143-f005:**
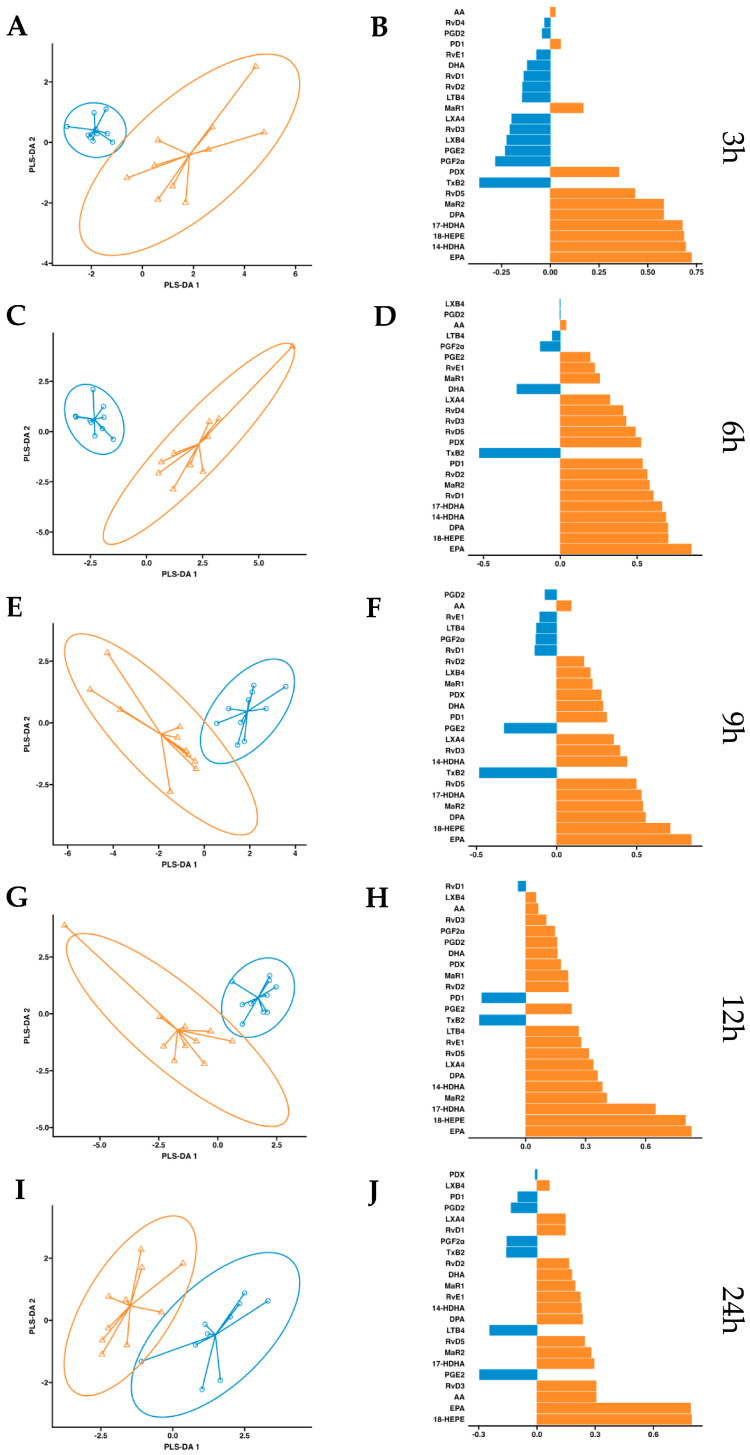
Plasma PLS-DA analysis. Results of partial least squares discriminant analysis in plasma. Left panel shows the sample plots from sPLS-DA, including 95% confidence ellipses (
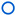
, control; 
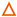
, supplemented). Right panel plots display the lipid mediator variable importance from component 1; colors indicate the group in which the median is maximum for each analyte (
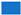
, control; 
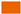
, supplemented). Different panels represent the results obtained at 3 h (**A**,**B**), 6 h (**C**,**D**), 9 h (**E**,**F**), 12 h (**G**,**H**) and 24 h (**I**,**J**) post-intervention.

**Figure 6 ijms-24-16143-f006:**
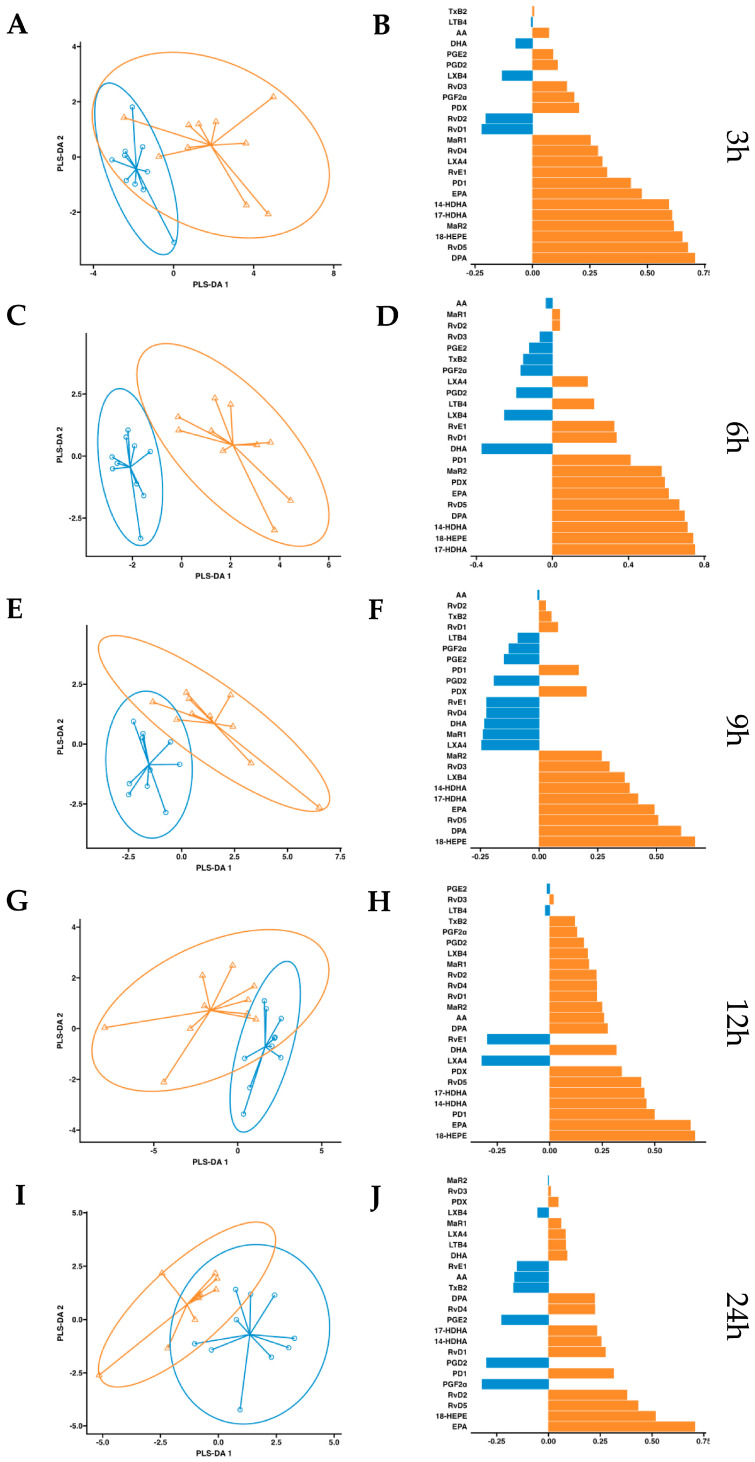
Serum PLS-DA analysis. Results of partial least squares discriminant analysis in serum. Left panel shows the sample plots from sPLS-DA, including 95% confidence ellipses (
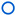
, control; 
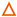
, supplemented). Right panel plots display the lipid mediator variable importance from component 1; colors indicate the group in which the median is maximum for each analyte (
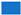
, control; 
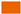
, supplemented). Different panels represent the results obtained at 3 h (**A**,**B**), 6 h (**C**,**D**), 9 h (**E**,**F**), 12 h (**G**,**H**) and 24 h (**I**,**J**) post-intervention.

**Table 1 ijms-24-16143-t001:** Demographic and baseline information of study participants.

Parameter	Values	Normal Values		
Participants	10			
Gender (male/female)	5/5			
Age (years)	31.81 ± 5.95			
BMI (kg/m^2^)	23.94 ± 3.41			
Glucose (mg/dL)	89.60 ± 5.42	74–106		
BUN (g/L)	0.33 ± 0.04	Male: 0.19–0.44 Female: 0.15–0.40		
Creatinine (mg/dL)	0.83 ± 0.11	Male: 0.7–1.2 Female: 0.5–0.9		
Albumin (g/dL)	4.19 ± 0.30	3.5–5.2		
Total cholesterol (mg/dL)	188.50 ± 23.70	National Education Cholesterol Program Adult Treatment Panel III No risk < 200 Moderate risk 200–239 High risk > 240		
c-HDL (mg/dL)	60.4 ± 17.00	National Education Cholesterol Program Adult Treatment Panel III At risk < 40 Negative risk > 60		
Non-HDL(mg/dL)	128.1 ± 26.8			
c-LDL (mg/dL)	109.78 ± 22.99	2019 ESC/EAS Guidelines for the Management of Dyslipidemias (Cardiovascular Risk)		
		Risk level	c-LDL	Non-c-HDL
		Very high	<55	<85
		High	<70	<100
		Moderate	<100	<130
		Low	<116	not apply
Triglycerides (mg/dL)	91.60 ± 46.56	National Education Cholesterol Program Adult Treatment Panel III Normal < 150 Borderline 150–199 High 200–499 Very high > 500		
AST (U/L)	21.50 ± 4.09	Male: 0–37 Female: 0–31		
ALT (U/L)	16.20 ± 4.69	Male: 0–41 Female: 0–33		
GammaGT (U/L)	16.20 ± 5.41	Male: 0–60 Female: 0–40		
Alkaline phosphatase (U/L)	57.20 ± 11.33	Male: 40–130 Female: 35–104		
Leukocyte count (mil/mm^3^)	6.56 ± 1.84	4–11		
Neutrophil count (mil/mm^3^)	3.22 ± 0.72	1.6–7.0		
Lymphocyte count (mil/mm^3^)	2.54 ± 1.37	1.2–1.4		
Monocyte count (mil/mm^3^)	0.47 ± 0.14	0.1–0.8		
Eosinophils count (mil/mm^3^)	0.21 ± 0.07	0.0–0.4		
Basophils count (mil/mm^3^)	0.04 ± 0.05	0.0–0.2		
Erythrocytes count (mill/mm^3^)	4.98 ± 0.32	Male: 4.5–5.9 Female: 3.5–5.1		
Hemoglobin (g/dL)	17.77 ± 1.18	Male: 13.0–17.4 Female: 12.0–15.3		
Hematocrit (%)	43.33 ±3.37	Male: 41.5–50.4 Female: 36.0–45.0		
MCV (fl)	86.91 ± 3.88	82–98		
MCH (pg)	29.63 ± 1.43	Male: 27–32 Female: 27–31		
Sedimentation rate, 1 h (mm)	7.10 ± 5.13	0–15		
Platelets count (mil/mm^3^)	244.30 ± 49.18	150–400		
Prothrombin time (s)	12.79 ± 0.60	9.8–14.6		
Prothrombin activity (%)	93.00 ± 7.18	80–120		
INR	1.05 ± 0.05	0.8–1.2		
Fibrinogen (mg/dL)	337.20 ± 47.82	200–400		

Values are expressed as mean ± standard deviation. BMI, body mass index; BUN, blood urea nitrogen; c-HDL, high-density lipoprotein cholesterol; c-LDL, low-density lipoprotein cholesterol; AST, aspartate aminotransferase; ALT, alanine aminotransferase; GammaGT, gamma-glutamyl transferase; MCV, mean corpuscular volume; MCH, mean corpuscular hemoglobin; INR, international normalized ratio.

**Table 2 ijms-24-16143-t002:** Pharmacokinetic parameters in plasma.

	Cmax							Tmax							AUC						
	Control			SPM-Enriched Marine Oil			*p*-Value	Control			SPM-Enriched Marine Oil			*p*-Value	Control			SPM-Enriched Marine Oil			*p*-Value
	AM	GM	SE	AM	GM	SE		AM	GM	SE	AM	GM	SE		AM	GM	SE	AM	GM	SE	
DHA metabolome																					
DHA	17.29	0.00	5.07	17.21	0.00	5.53	1.000	9.43	7.75	2.65	9.75	8.31	2.25	0.766	227.08	0.00	79.71	158.70	0.00	57.58	0.571
14-HDHA	1.25	0.00	0.87	31.51	27.53	5.02	**0.000**	13.00	10.69	3.61	5.40	5.22	0.40	**0.021**	10.67	0.00	6.47	181.70	153.53	33.73	**0.000**
17-HDHA	0.24	0.00	0.05	31.59	26.01	5.82	**0.000**	6.00	5.31	1.13	5.10	4.87	0.46	0.727	1.56	0.00	0.49	125.38	106.76	21.44	**0.000**
RvD1	6.75	0.00	4.54	5.08	4.61	0.75	**0.038**	7.50	5.58	2.54	7.20	5.83	1.96	0.888	37.61	0.000	14.53	62.89	44.94	16.43	0.602
RvD2	5.38	0.00	2.97	5.42	0.00	1.47	0.307	9.38	7.14	2.57	6.67	6.08	0.97	0.619	40.72	0.00	13.35	48.521	0.00	14.25	0.347
RvD3	5.69	0.00	4.94	1.83	1.69	0.24	0.064	10.71	7.44	3.58	9.00	7.56	1.90	0.960	20.68	0.00	14.60	15.47	12.80	2.763	0.667
RvD4	2.51	0.00	2.51	2.44	0.00	0.88	0.113	3.00	3.00	NA	3.60	3.45	0.60	1.000	7.52	0.00	7.52	9.80	0.00	3.73	0.779
RvD5	13.45	0.00	8.04	95.24	0.00	35.81	**0.013**	6.00	5.28	1.00	4.33	4.08	0.53	0.249	65.16	0.00	21.97	407.95	0.00	129.36	**0.007**
PDX	11.58	4.44	7.87	36.68	0.00	11.36	0.052	11.10	7.69	2.97	4.67	4.41	0.53	0.226	56.60	32.22	23.82	172.44	0.00	46.80	**0.011**
PD1	8.23	0.00	5.51	7.09	0.00	1.35	0.112	12.33	8.82	3.14	3.67	3.50	0.44	**0.028**	50.63	0.00	22.45	52.04	0.00	12.71	0.832
MaR1	18.23	0.00	6.46	41.78	0.00	16.37	0.121	8.25	6.31	2.52	7.67	6.08	2.19	0.960	142.42	0.00	44.72	403.09	0.00	195.59	0.102
MaR2	7.50	0.00	5.49	62.05	47.34	14.77	**0.001**	10.88	7.90	3.10	4.20	3.96	0.49	0.077	59.60	0.00	40.16	261.43	211.55	56.24	**0.000**
EPA metabolome																					
EPA	12.24	0.00	4.30	81.21	76.96	8.34	**0.000**	5.57	5.22	0.78	6.30	5.83	0.83	0.659	106.07	0.00	42.34	1182.33	1098.88	142.98	**0.000**
18-HEPE	0.18	0.00	0.06	41.33	33.53	8.66	**0.000**	8.67	6.78	2.26	4.20	3.96	0.49	0.087	0.93	0.00	0.24	178.36	151.26	34.06	**0.000**
RvE1	31.66	0.00	22.02	39.44	0.00	25.17	0.485	6.00	5.20	3.00	12.75	11.17	3.95	0.348	187.84	0.00	153.12	350.11	0.00	295.95	0.409
AA metabolome																					
AA	20.15	0.00	5.62	10.38	0.00	3.84	0.197	6.67	5.89	1.09	13.71	10.87	3.64	0.228	214.24	0.00	74.16	115.69	0.00	50.26	0.109
LXA_4_	6.48	0.00	5.41	2.03	1.71	0.40	0.241	9.38	6.00	3.38	6.00	5.08	1.18	0.922	25.12	0.00	15.76	18.42	14.24	3.76	0.619
LXB_4_	8.00	0.00	5.91	3.53	0.00	1.52	1.000	10.50	7.90	4.66	6.75	6.18	1.44	0.881	34.32	0.00	19.98	25.06	0.00	12.92	0.531
LTB4	109.08	97.76	17.84	95.62	78.64	19.65	0.631	14.10	11.01	2.83	13.20	10.88	2.54	0.969	1083.07	986.61	155.18	1147.52	812.18	271.20	0.511
PGD_2_	11.53	0.00	7.12	10.32	0.00	7.03	0.622	7.88	6.09	2.47	9.00	6.00	3.31	0.781	45.29	0.00	19.23	56.17	0.00	28.24	0.590
PGE_2_	31.52	15.69	8.97	32.03	21.30	11.91	0.912	9.60	7.26	2.52	6.00	5.29	1.00	0.410	135.04	77.29	30.82	302.23	183.77	93.72	0.101
PGF_2α_	17.35	0.00	7.86	12.43	0.00	5.94	0.791	5.63	4.70	1.32	8.25	6.00	2.65	0.523	108.11	0.00	41.47	114.30	0.00	57.91	0.980
TXB_2_	384.95	280.81	107.40	166.63	132.30	27.13	0.315	6.90	5.90	1.19	8.70	6.43	2.21	0.658	2913.55	1824.43	1031.20	1628.87	1209.08	323.46	0.196
DPA metabolome																					
DPA	27.96	0.00	7.99	131.44	124.19	12.46	**0.000**	6.86	6.63	0.86	5.40	5.22	0.40	0.123	283.94	0.00	85.06	885.68	834.56	109.40	**0.000**

AUC, area under the curve; AM, arithmetic mean; GM, geometric mean; SE, standard error. Statistically significant differences (*p* < 0.05) are marked in bold.

**Table 3 ijms-24-16143-t003:** Pharmacokinetic parameters in serum.

	Cmax							Tmax							AUC						
	Control			SPM-Enriched Marine Oil			*p*-Value	Control			SPM-Enriched Marine Oil			*p*-Value	Control			SPM-Enriched Marine Oil			*p*-Value
	AM	GM	SE	AM	GM	SE		AM	GM	SE	AM	GM	SE		AM	GM	SE	AM	GM	SE	
DHA metabolome																					
DHA	26.41	11.39	9.83	25.29	0.00	6.23	0.631	10.20	8.34	2.38	11.00	8.94	2.60	0.799	192.35	64.20	77.99	181.63	0.00	43.62	0.742
14-HDHA	27.33	0.00	10.01	142.20	123.93	24.68	**0.000**	9.67	7.56	2.73	6.90	5.60	1.95	0.295	229.37	0.00	100.79	1019.52	800.94	196.99	**0.000**
17-HDHA	4.31	0.00	1.73	63.80	0.00	13.35	**0.002**	10.50	8.19	3.00	5.33	5.14	0.44	0.127	36.33	0.00	17.10	328.22	0.00	65.23	**0.000**
RvD1	1.72	0.00	0.79	2.11	0.00	0.91	0.765	9.00	6.00	5.05	6.75	6.00	1.89	0.879	6.15	0.00	2.78	27.96	0.00	13.54	0.147
RvD2	3.82	0.00	2.02	4.76	0.00	1.12	0.184	9.00	7.51	2.34	9.38	7.62	2.37	0.829	27.30	0.00	10.63	48.91	0.00	12.94	**0.049**
RvD3	1.38	0.00	0.56	1.45	0.00	0.57	0.781	12.00	8.49	4.03	5.40	4.55	1.75	0.289	15.67	0.00	8.58	17.28	0.00	9.49	0.572
RvD4	0.29	0.00	0.21	4.96	0.00	2.76	0.121	6.00	5.20	3.00	4.80	3.96	1.80	0.809	0.88	0.00	0.64	17.95	0.00	9.79	0.131
RvD5	24.33	0.00	7.32	273.67	132.61	58.45	**0.001**	11.67	8.54	3.14	5.10	4.87	0.46	0.129	197.61	0.00	72.63	1524.07	665.60	332.83	**0.000**
PDX	37.18	0.00	15.43	98.85	85.28	16.49	**0.009**	15.38	12.62	3.28	7.20	6.00	1.91	**0.023**	519.84	0.00	327.01	797.49	610.21	169.81	**0.007**
PD1	4.64	0.00	2.16	11.68	8.64	2.74	**0.031**	7.50	5.31	2.66	6.60	5.22	1.99	0.848	27.62	0.00	12.42	78.03	53.33	19.43	**0.015**
MaR1	59.75	28.18	32.54	66.62	0.00	35.58	0.579	7.20	6.51	1.02	6.00	5.38	1.00	0.414	501.50	203.70	320.76	754.21	0.00	506.62	0.546
MaR2	38.14	0.00	13.20	249.45	0.00	77.37	**0.016**	10.00	7.56	2.78	5.67	5.56	0.33	0.460	380.39	0.00	155.03	1187.84	0.00	343.44	**0.001**
EPA metabolome																					
EPA	17.88	0.00	4.64	93.60	59.28	23.43	**0.002**	9.86	8.21	2.60	8.40	7.26	1.83	0.572	143.12	0.00	40.38	1078.94	635.09	248.02	**0.000**
18-HEPE	1.41	0.00	0.45	48.23	0.00	10.85	**0.002**	13.71	10.43	3.70	4.33	4.08	0.53	**0.022**	15.92	0.00	5.47	326.87	0.00	56.79	**0.000**
RvE1	103.61	0.00	66.11	40.76	0.00	14.82	0.532	12.60	10.45	3.34	9.43	6.00	3.79	0.393	708.14	0.00	422.16	365.62	0.00	170.34	0.296
AA metabolome																					
AA	34.00	0.00	10.87	26.18	0.00	9.51	0.545	13.67	11.78	2.65	10.00	8.94	1.94	0.336	284.67	0.00	95.71	172.00	0.00	39.99	0.306
LXA_4_	2.55	0.00	1.08	1.48	0.00	0.38	0.702	12.00	9.07	3.34	12.86	8.92	3.97	1.000	18.48	0.00	7.07	15.12	0.00	5.31	0.157
LXB_4_	6.39	0.00	4.11	10.88	0.00	5.43	0.482	11.25	8.49	4.64	13.80	10.70	4.31	0.802	41.09	0.00	27.45	70.86	0.00	30.28	0.509
LTB4	219.86	0.00	91.14	224.27	0.00	66.01	0.448	9.00	7.02	2.78	11.33	8.17	3.19	0.868	2271.84	0.00	1062.66	1962.96	0.00	643.55	0.676
PGD_2_	166.23	102.59	50.86	50.67	0.00	34.54	**0.014**	14.70	12.65	2.59	8.57	6.36	2.89	0.099	1449.61	772.28	478.45	523.32	0.00	336.81	**0.018**
PGE_2_	21848.97	0.00	6352.51	9748.40	0.00	6561.40	0.072	13.88	11.17	3.10	7.00	4.76	3.44	0.071	237403.15	0.00	85313.21	78700.96	0.00	63805.26	0.072
PGF_2α_	3559.09	0.00	1187.05	965.85	0.00	555.45	0.052	15.33	13.13	2.80	6.50	5.72	1.43	**0.040**	36333.89	0.00	13191.80	10065.52	0.00	7378.03	**0.021**
TXB_2_	24746.36	0.00	5917.52	29064.68	0.00	16808.12	0.307	12.67	9.96	2.95	13.50	10.24	3.26	0.882	321567.88	0.00	97455.88	250355.73	0.00	133126.57	0.577
DPA metabolome																					
DPA	28.80	0.00	7.54	111.40	0.00	22.95	**0.003**	9.00	7.31	2.70	5.00	4.76	0.50	0.158	233.06	0.00	78.44	724.19	0.00	165.43	**0.000**

AUC, area under the curve; AM, arithmetic mean; GM, geometric mean; SE, standard error. Statistically significant differences (*p* < 0.05) are marked in bold.

**Table 4 ijms-24-16143-t004:** Nutritional compositions of the food provided during this study.

	kcal	Fat (g)	Protein (g)	Carbohydrates (g)
Breakfast				
Skim milk, 1 cup	70.0	0.6	6.4	9.6
Bread, 2 slices with tomato	158.0	0.8	3.9	13.3
Apple, 1 medium	108.2	0.7	0.6	22.8
Lunch				
Lentils with vegetables	365.0	8.3	21.0	43.8
Grilled chicken	217.5	9.3	33.3	0.0
Pear, 1 medium	84.0	0.2	0.73	3.7
Snack				
Skim milk, 1 cup	70.0	0.6	6.4	9.6
Biscuits, 8 units	248.0	10.4	3.2	35.2
Apple, 1 medium	108.2	0.7	0.6	22.8
Dinner				
Salad	131.7	10.8	2.3	4.9
Grilled pork loin	466.5	39.9	24.4	1.5
Roasted green pepper	35.4	1.4	1.1	2.9
Low-fat yogurt	50.0	0.5	4.8	5.6

**Table 5 ijms-24-16143-t005:** Lower limits of quantification for specialized pro-resolving lipid mediators.

Analyte	LLOQ (pg/mL)
Docosahexaenoic acid, DHA	0.02
Docosapentaenoic acid, DPA	0.15
Eicosapentaenoic acid, EPA	0.01
Arachidonic acid, AA	0.02
14-hydroxydocosahexaenoic acid, 14-HDHA	0.08
17-hydroxydocosahexaenoic acid, 17-HDHA	0.33
18-hydroxyeicosapentaenoic acid, 18-HEPE	0.24
Leucotriene B4, LTB4	0.25
Lipoxin A4, LXA_4_	0.28
Lipoxin B4, LXB_4_	0.25
Maresin 1, MaR1	0.74
Maresin 2, MaR2	0.18
Prostaglandin D_2_, PGD_2_	0.17
Prostaglandin E_2_, PGE_2_	0.30
Prostaglandin F_2α_, PGF_2α_	0.43
Protectin D1, PD1	0.15
Protectin DX, PDX	0.30
Resolvin E1, RvE1	0.21
Resolvin D1, RvD1	0.64
Resolvin D2, RvD2	0.97
Resolvin D3, RvD3	0.30
Resolvin D4, RvD4	0.42
Resolvin D5, RvD5	0.25
Thromboxane B_2_, TXB_2_	0.29

LLOQs were determined by a signal-to-noise ratio of 5:1.

## Data Availability

The data supporting the present study are available in the article or will be obtained from the corresponding author upon request.

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
