# Peer review of "Pharmacokinetics and Changes in Lipid Mediator Profiling after Consumption of Specialized Pro-Resolving Lipid-Mediator-Enriched Marine Oil in Healthy Subjects"

_ijms, 2023, doi:10.3390/ijms242216143_

Round 1

Reviewer 1 Report

Comments and Suggestions for Authors

The manuscript entitled as “A Time course study of lipid mediators after consumption of SPM-enriched marine oil in healthy subjects.” is a nice, well planned and need of present era in which authors discussed the beneficial effects of Omega-3 polyunsaturated fatty acids in health as well as provide evidence of temporal increases of SPMs that together support the idea that given oral supplementation is a good source of essential bioactive SPM.  However I have some comments to improve the quality of this manuscript.

-Please rewrite the abstract and fortify the

-English language should be improve before publication.

-Line 94-98 please remove or more materials method section.

-results section. Please move some supportive results to supplementary file.

-Please numbered the material and method section.

-Please make two subsections of Pharmacokinetic and statistical analyses

- how you have selected the number of human statistically.

-Line 659, please write a bit detail of R programming codes.

-please provide the map of  study area.

-please some more relevant and recent references.

Comments on the Quality of English Language

Minor English changes required.

Reviewer 2 Report

Comments and Suggestions for Authors

In this manuscript, the interconnection between intake of SPM-enriched oil and concentrations of the bioactive lipid mediators in peripheral blood were determined. The pharmacokinetic fundamentals were monitored also. These works are very interesting. The article was well-written. Some details can be more clarified as below,

1. Title can be re-named after careful consideration. SPM should give the full name. Abstract section, it can be re-organized logically.

2. The design, the authors set two groups, one is control, another is enriched-SPM marine oil supplementation. What about the regular marine oil?

3. The data and results were well-expressed, however, the resolution of some figures can be improved.

4. The format of the references should be uniform and followed strictly with the requirements of Journal.

Reviewer 3 Report

Comments and Suggestions for Authors

This is an important study that will contribute to the overall body of evidence in regarding to fish oil and inflammation. 

1) Sentences in line 366-368 is not very clear and needs to be rephrased. 

2) The discussion needs improvements or it should be combined with the results. The discussion in the current state reads more like an introduction where they just mentioned the literature in terms of the effects of marine oils but, not in terms of their findings and how it would align (agree/disagree) with similar studies. 

3) A list of abbreviations would be very helpful. 

Comments on the Quality of English Language

The english language quality in general was very goos and well understood.

Round 2

Reviewer 2 Report

Comments and Suggestions for Authors

The comments have been point-to-point responded, and this manuscript is acceptable and more reasonable. Based on the revision, it can be accepted in this Journal.